# Vasodilating Effects of Antispasmodic Agents and Their Cytotoxicity in Vascular Smooth Muscle Cells and Endothelial Cells—Potential Application in Microsurgery

**DOI:** 10.3390/ijms241310850

**Published:** 2023-06-29

**Authors:** Misato Ueda, Yasuki Hirayama, Haruo Ogawa, Tadashi Nomura, Hiroto Terashi, Shunsuke Sakakibara

**Affiliations:** 1Department of Plastic and Aesthetic Surgery, Kobe University Graduate School of Medicine, Kobe 650-0017, Japan; sakuramochi0v0@yahoo.co.jp (M.U.); surounin26@live.jp (Y.H.); tadnomu@med.kobe-u.ac.jp (T.N.); terashi@med.kobe-u.ac.jp (H.T.); 2Hyogo Prefectural Harima-Himeji General Medical Centre, Himeji 670-8560, Japan; yeraishan@yahoo.co.jp

**Keywords:** microvascular anastomosis, vasospasm, antispasmodic agents, smooth muscle cells, endothelial cells, cytotoxicity

## Abstract

This study aimed to elucidate the vasodilatory effects and cytotoxicity of various vasodilators used as antispasmodic agents during microsurgical anastomosis. Rat smooth muscle cells (RSMCs) and human coronary artery endothelial cells (HCAECs) were used to investigate the physiological concentrations and cytotoxicity of various vasodilators (lidocaine, papaverine, nitroglycerin, phentolamine, and orciprenaline). Using a wire myograph system, we determined the vasodilatory effects of each drug in rat abdominal aortic sections at the concentration resulting in maximal vasodilation as well as at the surrounding concentrations 10 min after administration. Maximal vasodilation effect 10 min after administration was achieved at the following concentrations: lidocaine, 35 mM; papaverine, 0.18 mM; nitroglycerin, 0.022 mM; phentolamine, 0.11 mM; olprinone, 0.004 mM. The IC_50_ for lidocaine, papaverine, and nitroglycerin was measured in rat abdominal aortic sections, as well as in RSMCs after 30 min and in HCAECs after 10 min. Phentolamine and olprinone showed no cytotoxicity towards RSMCs or HCAECs. The concentrations of the various drugs required to achieve vasodilation were lower than the reported clinical concentrations. Lidocaine, papaverine, and nitroglycerin showed cytotoxicity, even at lower concentrations than those reported clinically. Phentolamine and olprinone show antispasmodic effects without cytotoxicity, making them useful candidates for local administration as antispasmodics.

## 1. Introduction

Microsurgery-based free flap procedures are now commonplace techniques with high success rates ranging from 94.3% to 99.2% [1,2,3,4]. While these procedures have a high success rate, they rely extensively on microvascular anastomosis for blood supply to the flap. Any abnormalities in the anastomosed vessels can substantially affect the flap’s viability. Although intraoperative vasospasm is well known, it often occurs unpredictably and, if it persists, can lead to decreased blood flow, stasis, and thrombosis in the flap [5]. Ultimately, this may result in total loss or partial damage to the flap [6]. Local administration of antispasmodic agents is performed during the procedure to prevent and mitigate vasospasm, but the types of drugs and their optimal concentrations are still a matter of debate. Furthermore, the extent of damage to the vascular tissue by these drugs remains unclear.

The mechanism underlying vasospasm during the procedure is complex and can occur in response to various stimuli. This response includes direct manipulation of blood vessels, metabolic homeostasis, and inherent tendencies [7]. Mechanical stretching of blood vessels can induce muscular reactions and alter the membrane potential. Rapid changes in the membrane potential can induce smooth muscle cell contraction and vasoconstriction. These reactions often occur during the handling and preparation of pedicles. Proper management of vascular dissection and minimization of tension is crucial [8].

The cellular components of blood vessels play a significant role in these reactions. The arterial wall is anatomically structured into three layers: intima, media, and adventitia [9] (Figure 1). Endothelial cells (ECs) are important constituents of the intima and perform various functions. ECs secrete numerous bioactive substances that regulate vascular tone. The release of substances such as nitric oxide (NO), prostaglandin I_2_ (PGI_2_), and endothelium-derived relaxing factor (EDRF), also known as an endothelium-derived hyperpolarizing factor (EDHF), induces local smooth muscle relaxation and subsequent vasodilation. In contrast, paracrine factors such as endothelin-1 and thromboxane A_2_ induce vasoconstriction. ECs also maintain the smoothness of the intima, prevent platelet and leukocyte adhesion, and protect against the entry of harmful molecules into the arterial vessel wall [10].

The media of blood vessel walls consists of multiple layers of smooth muscle cells (SMCs) [11]. SMCs are essential for proper blood vessel function [12]. Contraction and relaxation of SMCs cause changes in the diameter of the vessel lumen, enabling the blood vessel to maintain appropriate blood pressure. In addition to local factors, systemic hormones, such as angiotensin II and vasopressin, and sympathetic neurotransmitters, such as norepinephrine, increase the intracellular calcium ion concentration in SMCs by activating membrane-bound calcium channels through specific receptors, leading to vasoconstriction [13] (Figure 2). SMC function significantly affects the state of blood vessels during microsurgery.

Impairment of these cells can result in vascular tissue degeneration, disruption of vascular tone regulation, and potential thrombus formation. Consequently, flap survival remains a challenge. Local antispasmodic agents are administered during microsurgery; however, the optimal concentrations of antispasmodic drugs for effectively treating vascular spasms have not been established. Additionally, their effects on blood vessels remain unclear. Moreover, the effect and toxicity of these agents on physiological functions, as well as the optimal concentrations for use, is yet to be elucidated. The antispasmodic agents investigated in this study are conventionally used as injectable medications in clinical settings, where they are significantly diluted in the bloodstream, minimizing toxicity concerns. However, when used in microsurgery, they are directly applied to blood vessels and do not undergo dilution in body fluids, potentially exerting their effects at high concentrations. Consequently, there is a potential risk of inducing cellular toxicity. Therefore, we assessed the effects of different antispasmodic agents and their concentrations on blood vessels in rat abdominal aorta using a wire myograph system to quantify the contractile force of vascular smooth muscle and assess changes in vascular contractility during drug administration. Therefore, we elucidated the cellular toxicity of antispasmodic agents in vascular smooth muscle cells (VSMC) and human coronary artery endothelial cells (HCAEC), as well as their influence on vascular function.

## 2. Results

### 2.1. Quantification of Concentration-Vasodilation Relationship Using the Wire Myograph System

Consistent with our previous study [14], we applied a 1 gram-force (gf) load to approximately 1 mm cross-sectional slices of the rat abdominal aorta. Subsequently, we induced contraction by adding 5 μM norepinephrine. Various vasodilators were then added, and the concentration at which the vessels relaxed back to 1 gf was considered the concentration required for 100% relaxation [13]. Lidocaine hydrochloride, which is widely used in clinical practice, demonstrated approximately 100% relaxation at 1% lidocaine for 10 min. Therefore, we set an exposure time of 10 min to measure drug potency.

Figure 3a illustrates the relationship between the dilution level of commercially available formulations and corresponding relaxation rates (Appendix A). Figure 3b shows the relationship between drug concentration (mM) and relaxation rate. All tested agents, including lidocaine, papaverine, phentolamine, olprinone, and nitroglycerin, exhibited concentration-dependent relaxation effects. Lidocaine, papaverine, phentolamine, and olprinone demonstrated approximately 100% relaxation at concentrations of 35 mM, 0.18 mM, 0.004 mM, and 0.11 mM, respectively. However, nitroglycerin achieved only approximately 50% relaxation at the investigated concentrations.

Based on these findings, we prepared 11 concentrations (Table 1) of each agent, including concentrations near the 100% relaxation rate, in addition to a negative control. We examined the cytotoxicity of each concentration in RSMCs and HCAECs.

### 2.2. Measurement of IC_50_ Values for Each Drug

The IC_50_ values of each drug, as determined by a real-time cell viability assay, are presented in Table 2 and Table 3 (Appendix A). Lidocaine and papaverine displayed cellular toxicity even at low concentrations, as indicated by the findings at the 1-min post-exposure time point (Appendix A). In contrast, phentolamine and olprinone did not exhibit cellular toxicity at any concentration. Furthermore, the detection signals did not decline over time following exposure to phentolamine and olprinone in RSMCs and HCAECs (Appendix A).

#### 2.2.1. Lidocaine

The IC_50_ values at the 1-min time point post-administration were 55.3 mM for RSMCs and 24.4 mM for HCAECs. At 5 min post-administration, the IC_50_ values were 27.9 mM and 16.8 mM, respectively, while at 10 min, they were 16.7 mM and 13.5 mM, respectively. Subsequently, IC_50_ values decreased over time (Appendix A).

#### 2.2.2. Papaverine

At the 1-min time point post-administration, IC_50_ values were 0.10 mM for RSMCs and 0.056 mM for HCAECs. At the 60 min time point, the IC_50_ value was significantly lower at 0.0015 mM (Appendix A).

#### 2.2.3. Nitroglycerine

IC_50_ values were measured for RSMCs after 30 min of administration and for HCAECs after 10 min. The values were 0.13 mM for RSMCs (30 min later) and 0.035 mM for HCAECs (10 min later) (Appendix A).

#### 2.2.4. Phentolamine

IC_50_ measurements were conducted for RSMCs and HCAECs for up to 60 min. No cellular toxicity was observed. Cell viability was maintained, and no signal attenuation was observed (Appendix A).

#### 2.2.5. Olprinone

IC_50_ measurements were conducted for both RSMCs and HCAECs for up to 60 min, similar to that of phentolamine. No cellular toxicity was observed at any of the tested concentrations (Appendix A).

## 3. Discussion

During free flap surgery, vascular spasms can develop during vascular anastomosis. The frequency of this phenomenon is reported to be approximately 5–10% [15] in microscopic procedures. Vascular spasms can be induced by various stimuli, including direct manipulation of vessels during surgery, the action of vasoactive substances or hormones (related to metabolic homeostasis), and intrinsic characteristics [5,7]. However, the precise mechanisms underlying vascular spasms during microsurgery remain unclear [5]. Vasoconstriction reduces blood flow to the flap. Blood stasis can also lead to thrombus formation in the anastomosed vessels and within the flap. If not resolved, irreversible damage, including partial or complete flap loss, may occur.

To address these spasms, intraoperative administration of drugs for the prevention and relief of spasms has been adopted. A survey of the use of antispasmodic agents in the United Kingdom revealed that 94% of facilities routinely use drugs intraoperatively. Furthermore, 99% of surgeons perform local administration, and 19% additionally flush the intravascular lumen. The reasons for administration were primarily empirical (based on experience) and habitual, accounting for 42% and 21%, respectively [16]. Although clinical reports commonly mention the use of papaverine, verapamil, and lidocaine [16], research on the corresponding vasodilatory effects of various agents in vivo has also been conducted.

In vitro studies have also investigated vasodilation [17]. Although many studies have explored concentration-dependent vasodilatory effects in vitro and in vivo, the physiological effects of drugs on vessels have not been investigated. In our study, we used wire myography to measure the vasodilatory effects of drugs and investigated their pharmacological effects on blood vessels. Additionally, it is necessary to assess the potential tissue damage during clinical use. To evaluate the effect of each drug on the vasculature, we investigated its cytotoxicity in Ecs and VSMC.

### 3.1. Lidocaine

Lidocaine is a local anesthetic used as an antiarrhythmic agent. Although the vasodilatory effect of lidocaine is well-known, its precise mechanism of action remains unclear. This vasodilatory effect may result from the inhibition of voltage-gated Na^+^ channels, leading to a decrease in intracellular Na^+^ and Ca^2+^ concentrations in VSMC [18]. Studies have suggested that stabilization of the cell membrane, prevention of depolarization, and prevention of calcium influx are the major mechanisms underlying vasodilation [19]. Furthermore, reports indicate that the effects are primarily mediated through endothelium-dependent effects on neighboring autonomic nerve cells [20,21]. In addition to membrane stabilization to prevent depolarization and Ca channel activation, lidocaine can directly induce the relaxation of SMCs [20]. Lidocaine acts as a local anesthetic by blocking voltage-gated Na^+^ channels in the neuronal cell membranes. Its antispasmodic effect occurs rapidly, within 45–90 s, and lasts for 10–20 min [22].

Various reports have discussed the optimal concentration for the antispasmodic effects of lidocaine. Lidocaine exhibits biphasic concentration-dependent responses in vascular reactions, where low concentrations (1.5 × 10^−5^ to 1.5 × 10^−3^ M) cause vasoconstriction and high concentrations (4.5 × 10^−3^ to 1.5 × 10^−2^ M) relieve NE-dependent contractions and increase blood flow [23,24]. Beekman et al. [21] reported that lidocaine at 12% concentration showed antispasmodic effects similar to that of 20% lidocaine in rat tail arteries but with lower toxicity. Therefore, the authors reported that 12% lidocaine was the optimal concentration. Additionally, the same group reported that 12% lidocaine was more effective than lower concentrations and had minimal local tissue toxicity [25].

Lidocaine administration at concentrations of 10% [26] and 20% [20,27] has also been reported to be effective for vasoconstriction in rats. In our experiments on epinephrine-induced vasoconstriction in rat abdominal aorta, we [14] reported the dose-dependent vasodilatory effect of lidocaine. However, at concentrations higher than 10%, re-administration of epinephrine did not cause vasoconstriction, indicating the potential for irreversible changes in arterial tissue at high concentrations. We also showed that 5% lidocaine showed the best vasodilation with minimal irreversible changes in the arterial tissue. The safety of systemic administration of lidocaine at concentrations exceeding 2% in humans is not well established; however, there is no standard value for local administration. Joseph et al. [7] reported that the intraoperative local administration of a 2% lidocaine solution did not result in flap loss or a higher return rate to the operating room than with papaverine administration during a shortage. Johnstone et al. [28], although based on a few case reports, noted that 4% lidocaine was administered locally during microvascular reconstruction in humans. They reported that the peak concentration measured in peripheral blood, 1.5 μg/mL, was below the demonstrated toxicity level of 5–10 μg/mL and did not exhibit systemic toxicity. In the same study, epinephrine-induced vasorelaxation was observed with 1% lidocaine in rat aorta. However, the possibility of toxicity during direct vascular administration of high concentrations of lidocaine has also been suggested [29]. In clinical practice, local administration of 1% [7] or 2% [8] is more common. We used 1% lidocaine at 35 mM and 2% lidocaine at 70 mM. In this study, the IC_50_ of lidocaine is 24.4 mM (0.70%) for EC and 55.3 mM (1.58%) for SMC, even at 1-min post-dose values. At concentrations above those used in clinical practice, direct administration into the vascular lumen may damage vascular endothelial cells and SMCs. As shown in Table 1, the concentration of lidocaine used was higher compared to other drugs. While osmotic pressure itself could potentially cause cellular damage, the osmotic pressure of 1% lidocaine is 282 mOsm/L, which is not significantly different from that of physiological saline (290 mOsm/L) or 5% glucose solution (252 mOsm/L). At this concentration, the observed effects are more likely attributed to the inherent toxicity of the drug rather than osmotic pressure. However, it is worth noting that concentrations higher than 2% may not only exhibit cellular toxicity but also manifest its effect via osmotic pressure. Therefore, we postulate that 1% lidocaine is the optimal concentration and balanced in terms of toxicity. However, exposure should be less than 1 min and exposure to the vascular cavity should be avoided because endothelial and smooth muscle cells are damaged, even at low concentrations.

### 3.2. Papaverine

Papaverine is a nonspecific phosphodiesterase (PDE) inhibitor that is widely used as an intraoperative antispasmodic agent [5,7]. PDE is an enzyme extensively distributed in various systemic organs, including the myocardium, vascular smooth muscle, platelets, liver, bronchi, and lungs. Papaverine inhibits cyclic nucleotide phosphodiesterase, leading to the accumulation of intracellular cyclic guanosine monophosphate and a decrease in the contractile activity of VSMC [17,18]. In addition, they may possess calcium channel antagonistic properties, offering further potential for vasodilation [30]. Smooth muscle relaxation is manifested by a dual action involving an increase in intracellular cAMP levels due to PDE inhibition and the inhibition of intracellular calcium influx.

Papaverine is widely reported to be effective against vascular spasms. Although various administration methods, including the intraluminal, topical, and perivascular routes, have been documented, an ideal administration method supported by conclusive evidence is currently lacking [8]. Swartz et al. [31] reported that papaverine administration effectively improves the potency of microvascular anastomoses, demonstrating a significant reduction in and partial prevention of vascular spasms. Hou et al. [20] reported the statistically useful preventive and resolving effects of 0.3% papaverine compared to physiological saline on vasospasm induced in the rat femoral artery.

Previous studies have reported that the effects of papaverine become evident within a short time frame of less than 1–5 min after local administration [20,32,33,34].

However, there have also been reports of its toxicity. The concentrations of clinically used papaverine are highly acidic (pH, 3–4.5), raising concerns regarding potential tissue toxicity [17]. Cooper et al. [35] investigated the biochemical and morphological consequences of exposing the internal mammary artery endothelium to papaverine and concluded that the intraluminal injection of papaverine and papaverine solution acidified with physiological saline damaged the internal mammary artery endothelium. Gao et al. [36] reported that papaverine induces apoptosis and damages vascular endothelial cells and SMCs in animal experiments. As acidified physiological saline did not induce apoptosis in their study, they suggested that the toxicity was attributed to papaverine alone.

A clinical concentration of 30 mg/mL of papaverine is used during vascular anastomosis. This corresponds to 80 mM papaverine. Our wire myograph study showed that even at a significantly lower concentration of 0.18 mM hydrochloride papaverine, relaxation effects on the rat aorta induced by epinephrine were observed 10 min later, indicating that antispasmodic effects can be achieved at considerably low concentrations. The IC_50_ of papaverine hydrochloride was 0.097 mM for SMCs and 0.056 mM for ECs at 1 min, indicating an imbalance between physiological pharmacological effects and cellular toxicity. The direct application (perfusion) of papaverine at reported concentrations in the vascular lumen may impair both SMCs and ECs.

### 3.3. Nitroglycerin

Nitroglycerin, an organic nitrate, releases NO and increases intracellular cyclic guanosine monophosphate levels, resulting in the relaxation of vascular smooth muscle.

In the context of free-flap breast reconstruction, the intraoperative local administration of nitroglycerin has no significant difference in flap loss compared with the use of papaverine [37]. In an in vitro model using human skeletal muscle vessels, the local administration of nitroglycerin was reported to be highly effective in inducing concentration-dependent relaxation following endothelin-1-induced vascular spasms. This effect was observed not only in arteries but also in veins [18]. Ding et al. [38] reported the efficacy of local nitroglycerin administration in preventing phenylephrine-induced radial artery spasms in human radial arteries. However, the local nitroglycerin administration during internal mammary artery graft harvesting has been shown to not cause a significant increase in blood flow compared to the control group [39]. As a local formulation for intraoperative use during microsurgery, a 1 mg/mL solution was reported to be ineffective in shortening the duration of the vascular spasm in a rat model [16].

In this study, although vasodilatory effects were observed in the wire myograph measurements, the vasodilatory action was less than that of other drugs. Furthermore, IC_50_ values were obtained in SMCs and ECs 10 min after administration, indicating cellular toxicity after prolonged exposure. While there is a possibility of impairment of both SMCs and ECs with the direct application (perfusion) into the vascular lumen, this is not a concern in actual microsurgery, as there is no prolonged exposure of the luminal surface to vasodilators. However, considering its limited vasodilatory effect and cellular toxicity after prolonged exposure, it is not considered a suitable candidate for use during vascular anastomosis.

### 3.4. Phentolamine

Phentolamine is a non-selective α-blocker that induces vasodilation by inhibiting the release of intracellular calcium. This vasodilatory effect is primarily attributed to its direct action on the vascular smooth muscles [40]. Owing to its potent α-blocking activity, it is used in preoperative and intraoperative blood pressure control for pheochromocytoma-induced hypertension. Huang and Li [34] reported the rapid and potent relaxation of rat testicular vasculature following local administration of phentolamine immediately after epinephrine-induced vasoconstriction. However, it did not affect non-epinephrine-induced vasoconstriction. Their study revealed the onset of the effect within 2 min, with a sustained effect for up to 20 min, surpassing the potency of papaverine and magnesium sulfate. Ma et al. [41] demonstrated a significantly shortened relaxation time from vasoconstriction in mechanically induced rat femoral arteriovenous preparations with 10 mg/mL phentolamine drops compared with physiological saline. They also reported superior antispasmodic effects of 10% magnesium sulfate and 12% lidocaine. Ruiz-Salmerón et al. [42] investigated the ability of 2.5 mg verapamil or 2.5 mg phentolamine administered after sheath insertion to prevent arterial spasm in patients undergoing transradial cardiac catheterization. They reported that phentolamine exhibited lower antispasmodic efficacy than verapamil but demonstrated effective vasodilation.

In this study, we observed the maximum vasodilatory effect of phentolamine at a concentration of 0.11 mM 10 min after administration. This concentration was considerably lower than that reported by Ruiz et al. [42]. Furthermore, among the investigated agents, phentolamine demonstrated the most pronounced effect at the lowest molar concentration, suggesting potent vasodilatory action. Additionally, no IC_50_ values were measured in SMCs and ECs at various tested concentrations, indicating the absence of cellular toxicity. Phentolamine is a viable adjunctive vasodilator for local administration during vascular anastomoses. Reports on the use of phentolamine for the management of vascular spasms are limited and warrant further investigations, including safety considerations.

### 3.5. Olprinone

Olprinone, a selective inhibitor of PDEⅢ, is used as a therapeutic agent for acute heart failure. It is speculated to induce vasodilation by increasing intracellular cAMP and decreasing Ca^2+^ levels [43]. However, the optimal dosage for local administration remains unclear. Reports suggest that it inhibits noradrenaline- and KCL-induced contractions in the rat aorta, as well as phenylephrine-induced contractions in the canine aorta and saphenous vein [44]. Adachi et al. [45] described potent endothelium-independent dilation in the human radial artery and reported dose-dependent relaxation of phenylephrine-contracted radial artery strips (vessels). Sasaki et al. [46] reported olprinone as an alternative for local vasodilation of the right gastroepiploic artery during cardiac bypass.

In this study, we observed significant vasodilatory effects of olprinone. Furthermore, no cellular toxicity was observed in ECs or SMCs at the tested concentrations. It is an antispasmodic agent and is locally administered during vascular anastomosis. There are only a few reports on the use of olprinone for vasospasms in the surgical field, necessitating further investigation.

All agents examined in this study demonstrated vasodilatory effects in vivo. Although various drugs at various concentrations have been used for local intraoperative administration during microsurgery, their optimal concentration and physiological effects have not yet been established. In this study, we investigated the optimal concentration of vasodilatory agents based on their physiological effects and cytotoxicity. Vasodilatory effects were observed at concentrations lower than those reported previously [14]. Cytotoxicity tests showed no cellular toxicity of phentolamine or olprinone in VSMC and ECs. Direct action on SMCs is necessary to induce vasodilation, which may require passage through the intima or adventitia. Typically, in the vessels used in microsurgery, not only the anastomotic sites but also the vascular pedicle of the skin flap may experience spasms. While the anastomotic site exposes the adventitia, towards the skin flap side, the connective tissue surrounds the blood vessels, hindering drug penetration. Therefore, administration through the vascular lumen is presumed to be more efficient for penetrating the media. However, as our data demonstrated, lidocaine and papaverine exhibited significant cytotoxicity at concentrations at which a 100% relaxation rate was achieved. Consequently, when using these drugs, it may be beneficial to avoid perfusion through the lumen and instead administer short-term exposure multiple times, such as by spraying a low-concentration agent followed by prompt rinsing.

In addition, we investigated the pricing of medications, considering medical and economic aspects. Table 4 summarizes the selling prices of drugs in Japan and the cost per concentration (per 10 mL of diluent) representing the maximum vasodilatory effect 10 min after administration into the rat abdominal aorta. Xylocaine injection polyamp 1% 5 mL (Sandoz, Tokyo, Japan) was priced at ¥59.00, papaverine hydrochloride injection 40 mg 4% 1 mL (Nichi-Iko, Toyama, Japan) at ¥94.00, milrinone injection 1 mg/2 mL (Nippon Kayaku Co., Tokyo, Japan) at ¥122.00, regitine injection 5 mg (Novartis Pharma, Tokyo, Japan) at ¥59.00, and Coatec injection 5 mg (Eisai Co., Tokyo, Japan) at ¥3003.00. From a medical and economic perspective, phentolamine, which demonstrates sufficient vasodilatory effects without observable cytotoxicity, is the most desirable vasodilatory agent. However, phentolamine is a potent α-blocker, and its systemic effects, when applied in the surgical field during vascular anastomosis, remain uncertain, necessitating further research. We believe that safer concentration settings can be achieved by combining this method with toxicity testing.

This study had several limitations. First, vasodilation and vasoconstriction are primarily mediated by SMCs. Many vasodilators available in the market are developed to act on SMCs. ECs, which are adjacent to SMCs, release various factors that contribute to their contraction and relaxation. However, their effects are indirect, and drugs that directly target ECs are not desirable in situations where rapid effects are expected, such as in microsurgery. Therefore, the investigation in this study did not consider drugs that depend on ECs. Second, the blood vessels used in the wire myograph system were the rat abdominal aorta, whereas the vessels anastomosed in the free-flap surgery were peripheral arterioles, which may have different physiological functions from those of the major arterial system. However, in this study, the relative value of the drug effects was considered, allowing for the possibility of comparing the potency of these drugs. In addition, the cultured cells used in the toxicity test were derived from peripheral blood vessels, which may provide information in a more clinically relevant environment. However, it should be noted that the ECs were of human origin, and the SMCs were from rats, as obtaining them from the same species presents practical challenges. Drug action and resistance may vary among animal species, and a comparison of toxicity among different cell types is not feasible. The thrombotic rate of endothelial cell necrosis caused by toxic drugs requires further investigation. However, considering our previous study, in which 100% potency was observed even with decellularized blood vessels in rats [47], rats may be relatively resistant to the effects of intimal damage, and further investigation using other animal models may be necessary.

## 4. Materials and Methods

### 4.1. Animals

In vivo experiments were conducted to pharmacologically analyze the vasodilatory effects of various antispasmodic drug concentrations in the rat abdominal aorta using a wire myograph system. This study was approved by the Institutional Animal Care and Use Committee (Approval Number: R111106-R1) and conducted in accordance with the guidelines of the Kobe University Animal Experimentation Regulations. Mature female Wistar rats (average body weight: 300 g) were purchased from CLEA Japan (Tokyo, Japan). Lidocaine (Combi-Blocks, San Diego, CA, USA), papaverine (Nichi-Iko Pharmaceutical Co., Toyama, Japan), nitroglycerin (Nippon Kayaku Co., Tokyo, Japan), phentolamine (Novartis Pharma K.K., Tokyo, Japan), and orciprenaline (Eisai Co., Tokyo, Japan) were used as antispasmodic agents. The investigation was conducted using concentrations that were feasible for dilution based on commercially available drugs considering their practical applications. All procedures were performed under anesthesia induced by intraperitoneal injections of midazolam and butorphanol tartrate. The animals were then euthanized by exsanguination.

### 4.2. Wire Myograph System

A wire myograph system (Bio Research Co., Osaka, Japan) was used, as previously reported by Ogawa et al. [14]. The rat abdominal aortae were cut into rings and manipulated using two L-shaped hooks. One hook was fixed, and the other was connected to an amplifier to measure the force exerted on the pulled vessel in grams. The rats were deeply anesthetized, and the abdominal aorta was dissected. The excised vessels were immersed in Ringer solution (NaCl 118 mM, KCl 4.7 mM, CaCl_2_/2H_2_O 2.5 mM, MgSO_4_/7H_2_O 1.2 mM, KH_2_PO_4_ 1.2 mM, NaHCO_3_ 25 mM, C_6_H_12_O_6_ 11 mM, Na_2_EDTA 0.5 mM). The rat aorta was then sliced into approximately 1 mm-wide rings and hooked onto L-shaped hooks. The chamber was filled with 3 mL of Krebs-Ringer solution, and a gas mixture of 95% O_2_ and 5% CO_2_ was bubbled through. The chamber was maintained at 37 °C.

Once the wire myograph system was initiated, the wire was moved, and a load was exerted on the hook-attached chamber. The applied tensile force, amplified by a transducer and PowerLab (ADInstruments, Bella Vista, New South Wales, Australia), was continuously recorded as gram force using LabChart (ADInstruments, Bella Vista, New South Wales, Australia). The hook is pulled to apply a load of up to 1 gf. A resting period of approximately 10 min was required for the gradual relaxation of the vessel under tension from the hook. When relaxation reached a plateau, norepinephrine was added to the solution to achieve a final concentration of 5 μM. This causes the vessel to contract, increasing the load on the artery and surpassing 1 gf on the myograph recording. After the values stabilized, various antispasmodic agents (excluding lidocaine) were added to the chamber solution, and the relaxation of the vessel was measured. We utilized the lidocaine data from our previous research findings [14]. The reduction in load on the vessel in terms of gf was assessed to determine the relaxation rate and concentration (Figure 4).

The cancellation of norepinephrine-induced vasoconstriction or the maximum relaxation rate of the vessel 10 min after the administration of each antispasmodic agent was compared. It should be noted that considering practical clinical applications, lidocaine presented challenges in dilution below 1% (35 mM) owing to product specifications, and this concentration was used as the reference point.

### 4.3. Cell Culture

Rat smooth muscle cells were cultured in DMED:F12 (Lonza, Besel, Switzerland) containing 10% fetal bovine serum, 1% penicillin/streptomycin solution (GA-1000; Lonza, Basel, Switzerland), and 1% stable glutamine (Wako, Tokyo, Japan) at 37 °C in a 5% CO_2_ incubator. The medium was changed every three days. Once the adherent cells reached 70–80% subconfluence, they were detached from the dish using Accutase (Nacalai Tesque, Kyoto, Japan). The cell suspension was diluted with DMED:F12 (+FBS) and adjusted to a concentration of 5 × 10^4^ cells/mL, and then 100 μL of the suspension was seeded into each well of a 96-well dish (Nunc™ MicroWell™ 96-Well, Nunclon Delta-Treated, Flat-Bottom Microplate, Thermo Fisher, Waltham, MA, USA) and incubated for two days at 37 °C in a 5% CO_2_ incubator. Passage 6 smooth muscle cells were used, and five 96-well plates were prepared.

Human coronary artery ECs were cultured in a collagen-coated dish (AGC TECHNO GLASS Co., Shizuoka, Japan) in EBM-2 (CC-3156, Lonza, Besel, Switzerland) containing Supplements and Growth Factors (CC-4176, Lonza, Besel, Switzerland). The cells were cultured at 37 °C in a 5% CO_2_ incubator. The medium was changed every three days until the adherent cell monolayer reached 70–80% subconfluence. Subsequently, cells were detached from the dish using Accutase. The cell suspension was diluted with medium, adjusted to a concentration of 5 × 10^4^ cells/mL, and seeded into a 96-well plate. The dish was incubated for two days at 37 °C in a 5% CO_2_ incubator. En ECs at passages 5–6 were used, and five 96-well plates were prepared.

### 4.4. Cell Viability Assay

IC_50_ is the drug concentration at which half of the cells undergo cell death; it serves as an indicator of cellular toxicity. Cell viability and IC_50_ values of various vasodilators (lidocaine, papaverine, nitroglycerin, phentolamine, and olprinone) were investigated using RSMCs and HCAECs. IC_50_ values were measured using the GloMax^®^ Navigator System (Promega, Madison, WI, USA). The test compounds were diluted in the culture medium and stored in an incubator until use. The concentration of each compound was determined based on its potency to induce maximum vasodilation in the rat abdominal aorta 10 min after administration, as measured using the wire myograph system described earlier. For each test compound, a control and 11 different concentrations were prepared, and the cytotoxicity after the addition of the compounds was examined. The concentration of each compound is listed in Table 1. The RealTime-GloTM MT Cell Viability Assay kit (Promega, cat#G9711, USA) was used, and the substrate and Nano-luc were added to the medium in each well of a 96-well plate seeded with RSMCs and HCAECs, with 50 μL per well. The plate was incubated for 1–3 h in the incubator. Then, 50 μL of each prepared test compound was added to the plate. The IC_50_ values were measured using GloMax at 1, 5, 10, 30, and 60 min after the addition of the test compounds, and cytotoxicity was evaluated.

## 5. Conclusions

Lidocaine, papaverine, and nitroglycerin exhibited vasodilatory effects in rat abdominal aorta at concentrations lower than those reported for clinical use for their antispasmodic effects. Moreover, in cell viability assays, these compounds were cytotoxic to SMCs and ECs at concentrations lower than those used clinically. During microsurgery, when blood flow in the vessels is interrupted, the drug concentrations directly affect the tissues without dilution or washout. Therefore, when using lidocaine, papaverine, or nitroglycerin to achieve antispasmodic effects during vascular anastomosis, it is advisable to limit their application to the surface of vessels rather than flushing the vessel lumen. In contrast, phentolamine and olprinone exhibit antispasmodic effects without cytotoxicity. Considering its cost-effectiveness, phentolamine may be a desirable antispasmodic agent and may be a fitting option for local administration during vascular anastomosis.

## Figures and Tables

**Figure 1 ijms-24-10850-f001:**
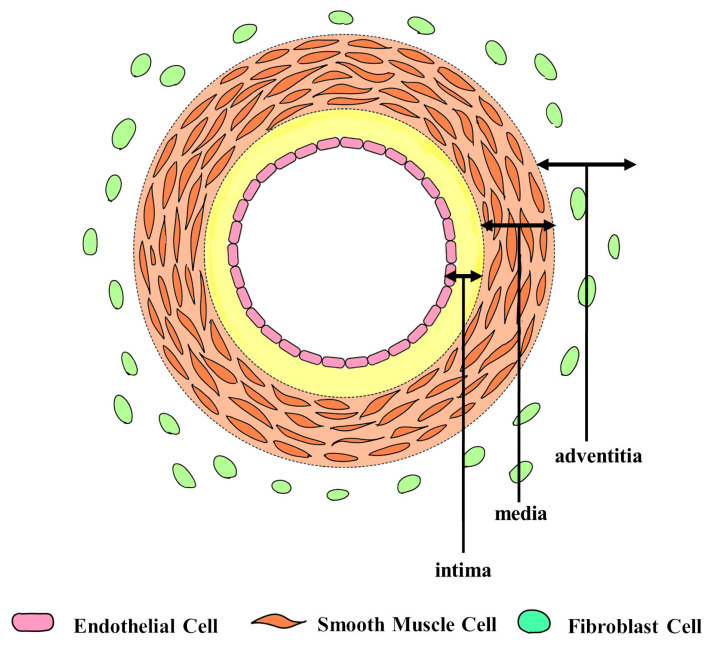
The blood vessel walls consist of three main layers: intima, media, and adventitia. Endothelial cells are crucial components of the intima. The media is made up of multiple layers of smooth muscle cells.

**Figure 2 ijms-24-10850-f002:**
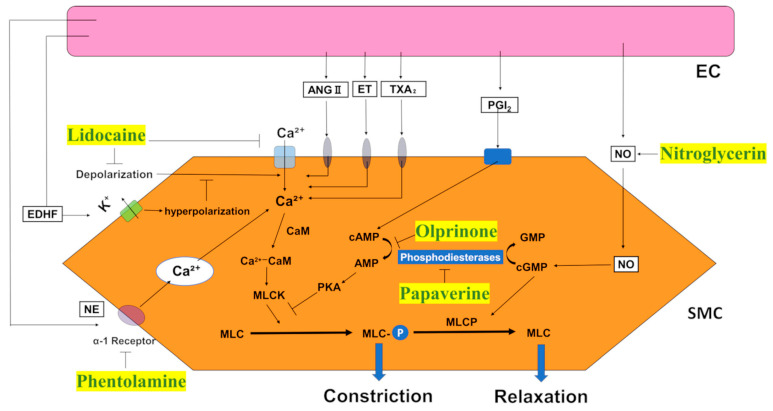
The mechanism of action of topical vasodilators on vascular smooth muscle cells. Vasodilators (green text with yellow background) act by either decreasing intracellular calcium concentrations or increasing nitric oxide, cyclic adenosine monophosphate, or cyclic guanosine monophosphate. EC, endothelial cell; SMC, smooth cell muscle; EDHF, endothelium-derived hyperpolarizing factor; ANGⅡ, angiotensinⅡ; ET, endothelin; TXA_2_, thromboxane A_2_; PGI_2_, Prostaglandin I_2_; NO, nitric oxide; NE, norepinephrine; Ca^2+^, calcium; cGMP, cyclic guanosine monophosphate; cAMP, cyclic adenosine monophosphate; GMP, guanosine monophosphate; AMP, adenosine monophosphate; CaM, Calmodulin; MLCK, myosin light chain kinase; MLC, myosin light chain; MLCP, myosin light-chain phosphatase.

**Figure 3 ijms-24-10850-f003:**
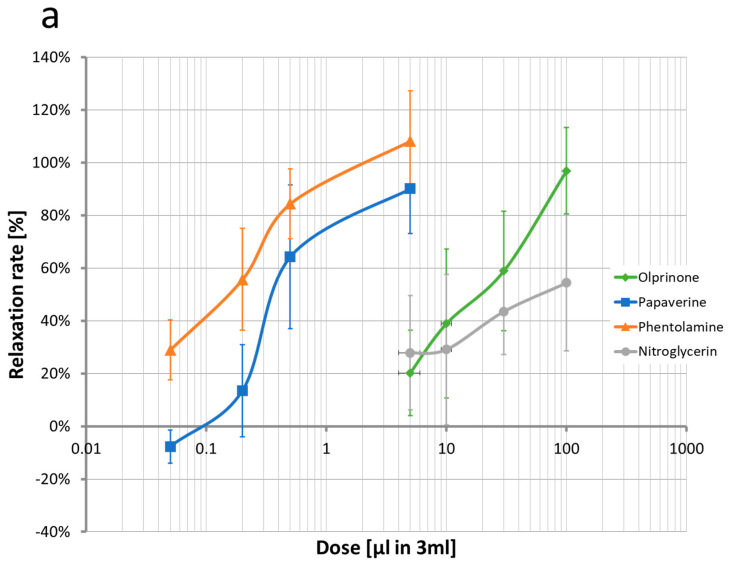
Relationship between vasodilator relaxation effects and concentration of each vasodilator. (**a**) The vascular relaxation rate at various drug doses. The potency of each drug is shown. 1% lidocaine (35 mM) was used as the standard and is excluded from this graph. (**b**) The vascular relaxation rate at various drug concentrations.

**Figure 4 ijms-24-10850-f004:**
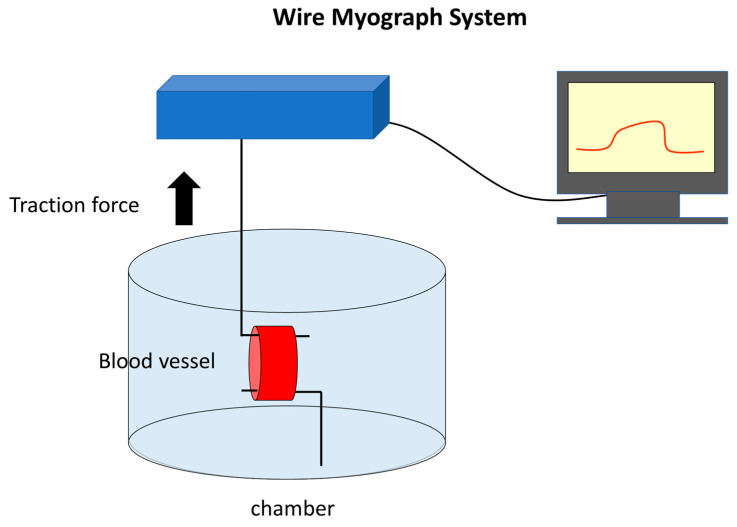
Wire myograph system.

**Table 1 ijms-24-10850-t001:** Concentration of each drug for measurement of cytotoxicity.

No.*	1	2	3	4	5	6	7	8	9	10	11	12	
Lidocaine	0	1.73 × 10^−1^	3.46 × 10^−1^	6.93 × 10^−1^	1.73	3.46	6.82	17.31	34.63	69.25	173.13	346.26	
Papaverine	0	3.55 × 10^−4^	7.10 × 10^−4^	1.77 × 10^−3^	3.55 × 10^−3^	7.10 × 10^−3^	1.77 × 10^−2^	3.55 × 10^−2^	7.10 × 10^−2^	1.78 × 10^−1^	3.55 × 10^−1^	7.10 × 10^−1^	
Nitroglycerine	0	1.10 × 10^−4^	2.20 × 10^−4^	5.50 × 10^−4^	1.10 × 10^−3^	2.20 × 10^−3^	5.50 × 10^−3^	1.10 × 10^−2^	2.20 × 10^−2^	5.50 × 10^−2^	1.10 × 10^−1^	2.20 × 10^−1^	
Phentolamine	0	4.42 × 10^−5^	8.83 × 10^−5^	2.21 × 10^−5^	4.42 × 10^−4^	8.83 × 10^−4^	2.21 × 10^−4^	4.42 × 10^−3^	8.83 × 10^−3^	2.21 × 10^−3^	4.42 × 10^−2^	8.83 × 10^−2^	
Olprinone	0	1.64 × 10^−4^	3.28 × 10^−4^	8.20 × 10^−4^	1.64 × 10^−3^	3.28 × 10^−3^	8.20 × 10^−3^	1.64 × 10^−2^	3.28 × 10^−2^	8.20 × 10^−2^	1.64 × 10^−1^	3.28 × 10^−1^	[mM]

* The numbers correspond to the rows of a 96-well plate.

**Table 2 ijms-24-10850-t002:** IC_50_ values of rat smooth muscle cells (RSMCs).

	1	5	10	30	60	(min)
Lidocaine	55.3121	27.9377	16.693	10.106	7.1017	
Papaverine	0.09704	0.11823	0.01348	0.0159	0.00146	
Nitroglycerine	ND	ND	ND	0.1339	0.1093	
Phentolamine	ND	ND	ND	ND	ND	
Olprinone	ND	ND	ND	ND	ND	[mM]

ND, not determined.

**Table 3 ijms-24-10850-t003:** IC_50_ values of human coronary artery endothelial cells (HCAECs).

	1	5	10	30	60	(min)
Lidocaine	24.4	16.8	13.5	10.3	7.64	
Papaverine	0.056	0.00404	0.00429	0.005	0.00648	
Nitroglycerine	ND	ND	0.0351	0.038	0.0388	
Phentolamine	ND	ND	ND	ND	ND	
Olprinone	ND	ND	ND	ND	ND	[mM]

ND, not determined.

**Table 4 ijms-24-10850-t004:** Cost of drugs of concentration showing maximal vasodilatory effect after 10 min in rat abdominal aorta per 10 mL of diluted solutions.

	Lidocaine	Papaverine	Nitroglycerin	Phentolamine	Olprinone
Cost (Yen)	118	1.59	4.99	0.18	201.32

## Data Availability

All data are available in the manuscript and within Appendix A.

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
