# Peer review of "Vasodilating Effects of Antispasmodic Agents and Their Cytotoxicity in Vascular Smooth Muscle Cells and Endothelial Cells—Potential Application in Microsurgery"

_ijms, 2023, doi:10.3390/ijms241310850_

Round 1
Reviewer 1 Report
Dear authors, please find attached the review report.

Minor improvements are necessary.
Reviewer 2 Report
The manuscript MS 2450925 “Vasodilating Effects of Antispasmodic Agents and their Cytotoxicity towards Vascular Smooth Muscle Cells and Endothelial Cells—Potential Application in Microsurgery" submitted to the International Journal for Molecular Sciences brings a description of vasodilatory effects and cytotoxicity of various vasodilators used as antispasmodic agents during microsurgical anastomosis. The manuscript is well-written and easy to follow. However, I have several major concerns listed in the comments.
General comments:
1. English proofreading is needed.
2. Add “50” in superscript in IC50 through text.
3. Why the authors didn’t use positive control for cytotoxicity assay?
4. I have concerns about applying a high concentration of lidocaine (over 35mM) because if it is applied to tissue/cells, the concentration itself can cause damage, not the compound itself. Add comments to the text.
Specific comments
1. In section 2.1. please write the meaning of “gf” because it is first mentioned here.
2. Mention what happens to the tested agents in the body after a certain time, are they metabolized, excreted or?
3. Explain why the relaxation of lidocaine is 130 % in Figure 3b, i.e. how can there be more than 100 % relaxation?
4. Explain what “*” represent in “No.*” in Table 1.
5. Table 1 is unreadable, make it in a smaller format so that the numbers are easy to read.
6. Explain how the IC50 values ​​were determined from the graphs in Figure S1-10 because the graphs are in RLU and are not sigmoidal in relation to the percentage of inhibition.
7. Please use smaller dots on the graphs in Figure S1-10 because the concentrations are not clearly visible.
8. Explain why RLU rises and then falls in Figure S1-10?
9. In line 260 dash is missing in “…highly acidic (pH 3 4.5)..”
10. In line from 266 the authors cite studies that showed that acidified medium does not cause apoptosis. It is important to emphasize that apoptosis is not the only “form” of death/toxicity and in my experience acidified medium affects the inhibition of cell growth.
11. Please mention Figure 4. in the text.
12. In section 4.3. Cell Culture, add “4” in superscript in “5 × 104 cells/ml”
13. Please correct through text “ml” in “mL”.
14. In section 4.3. Cell Culture add how many independent experiments with repetitions (duplicates or triplicates…) were done?
English proofreading is needed.
Round 2
Reviewer 1 Report
Dear authors, thank you for the revision of this interesting manuscript. I have no more remarks.
Nothing to add.
